# Informational Needs for Dental-Oriented Electronic Health Records from Dentists’ Perspectives

**DOI:** 10.3390/healthcare11020266

**Published:** 2023-01-14

**Authors:** Abdullah Alanazi, Ghada Alghamdi, Bakheet Aldosari

**Affiliations:** 1Health Informatics Department, King Saud Ibn Abdulaziz University for Health Sciences, Riyadh 11481, Saudi Arabia; 2King Abdullah International Medical Research Center, Riyadh 14611, Saudi Arabia

**Keywords:** electronic health records, dental health records, dentists’ informational needs

## Abstract

Introduction: Information technology is vital to support dental care services and is yet to be thoroughly investigated. This study aims to assess the dentists’ needs and requirements for health records from dental care providers’ perspectives. Methods: In-depth interviews were conducted with dentists during clinic practices. This qualitative research method involves exploring the information and functions dentists use to obtain information from EHR. The target population is the dental staff interacting with the patients and accessing the electronic health records in the government and private sectors. Results: Thirty-five dentists were interviewed directly after the treatment session and asked four pre-defined questions, the dentists’ needs were collected, and the met and unmet needs were presented. The interview results revealed 42 needs (15 were met and 27 were unmet), with an average of 1.17 needs per session. The information needs were categorized into foreground and background information needs and reported in nine main themes. Discussion: The interviews were analyzed, and as a result, nine themes were generated: chief complaints and symptoms, medical and health history, medications, visual representations of the problem, treatment procedures, X-ray services, needs related to advanced features, needs related to insurance coverage, and finally, information needs related to the treatment environment. The required information and functions mentioned by dentists in the study emphasize the need for integrated modules for oral and medical care services. Generally, it is evident that dentists have substantial unmet needs, and the desired EHR should have functions that cover all dentists’ needs. Conclusion: The study’s findings demonstrate gaps between current and desired EHR to serve dentists’ needs. Dentists need better access to patient history and medical information, progress notes, and X-rays to provide visualization tools for problems and patient charts. Moreover, essential needs were related to messaging capability, educational tools, availability of tutorial videos, and accessing external resources. Information needs were described and should be considered when designing EHR to meet all dentists’ needs.

## 1. Introduction

The electronic health record (EHR) is not merely a technology product but a means for clinical transformation aiming for clinical care excellence. This is only attainable by providing access to the most accurate, complete, and updated patient information. Such data would enable clinicians to make better decisions and communicate with other care providers, improving health outcomes and resource utilization [1]. A need has emerged for dental-oriented patient records aiming to enhance the quality of dental care by improving the accuracy and timeliness of clinical documentation and images and facilitating disease monitoring and control [2]. The advantages of a dental-oriented EHR are not limited to enhancing access to dental information and history, further improving communication with other dentists, continuity of care, and reducing narcotics abuse [3]. It is also essential for dental counseling and enabling rapid response in emergencies, especially for patients with complex medical conditions [4]. However, some challenges faced using EHR in dental clinics include the cost of the system, design, required functions, and implementation hassles, including transition and training [2].

### Background

The oral cavity is the window to overall health, considering that multiple systematic and chronic conditions such as diabetes or hypertension have oral manifestations. Oral health status can be a sign, as symptoms could appear in the oral cavity, or poor oral care may exacerbate the chronic conditions’ symptoms [5]. Early examination and planning intervention can enhance oral health and, for example, glycemic outcomes for diabetic patients. Hence, access to relevant health record systems can help medical professionals and dentists identify patients with a disease at risk of underdiagnosing [5]. 

Recently, best clinical practices and research findings emphasized the need to improve inter-professional coordination and collaboration, which brings value via exchanging patient data between dental and medical practices. High priority when designing health record systems goes for clinical tasks; hence, the plans are intended less on dental needs, including tooth and periodontal charts, images, and terminology [1]. The lack of dental consideration includes data previously entered digitally and on paper, such as tooth conditions, radiographs, insurance coverage, and authorization. Dentists have even indicated the capabilities of EHR to present better visual representations in addition to traditional radiographs and intraoral pictures, such as mounted case images and 3D models with the best image quality [6]. Unfortunately, a universal treatment plan is difficult to reach due to a lack of proper communication and different stakes, insurance coverage, and training approaches [7]. Additionally, clinical documentation and record-keeping standards are not the same among medical and dental domains, which needs resolutions to establish a universal documentation standard and ensure interoperability [1].

Dentists have made notable progress toward adopting EHRs despite the need for regulatory incentives. On average, a dental module in EHR systems costs USD 10,000–20,000, with additional training, maintenance, and support expenses. Costs associated with EHR adoption often exceed reported benefits by more than 23% [8]. Enhancements to EHR are usually overlooked, especially for pediatric dentistry, due to a lack of return on investment despite potential benefits [9]. Large dental practices see EHRs as vital for efficiency, while smaller practices, such as most dental providers, see it as costly and broadly disrupting workflow. Although dental-oriented EHR requires standardized tools for diagnosis and documentation, such as standardized dental diagnostic terminologies (SDDxTS) and Dental Diagnostic Systems (DDS), there are inherent barriers, such as poor system usability and a lack of scalability [10].

A significant challenge in designing dental-supported EHRs is incorporating the dentist’s evidence-based information requirements and seamlessly integrating the system into the workflow [6]. Dentists require various information from EHRs, such as chief complaints and symptoms, treatment options and procedures, and problem parameters. EHRs should be accessible due to their direct relation with dental conditions [6]. Therefore, these immediate needs of dentists should be the driver for designing integrated EHR systems.

Regarding the EHR, the required inputs based on a study by Sadoughi et al. are periodontal status and legal consent [11], while Shea et al. mentioned that the significant features requested in dental EHRs include oral health risk assessment, fluoride varnish applications, and dental referrals. This is particularly important for specializations such as pediatrics, where primary care physicians place significant importance on elements of oral health, nearly as important as traditional primary care information such as immunizations [9].

Although all the studies mentioned above indicate the need for dental–medical EHR, there still needs to be more robust evidence and guidance in implementing dental-oriented EHR. The American Dental Association has a list of recommended indices that should be included in dental records, but these recommendations must be updated as flaws exist [12]. A recent study demonstrated that around 30% of health information in dental clinics is misreported [5]. This study aims to identify the dental information necessary to deliver proper treatment from the dentist’s point of view, and furthermore, to underline the information required to be added to the electronic health record to improve the quality of dental services.

## 2. Materials and Methods

The study was conducted via in-depth interviews with dentists during clinic practices. This qualitative research method involves conducting a single intensive interview during the treatment sessions to understand dentists’ feelings, points of view, and feedback. Throughout the interview, the investigators asked and explored the missing information and tools dentists use to obtain information from EHRs. The questions of the study were adopted from a previous study [6]. Our target population was dentists interacting with patients and accessing electronic health records in Riyadh, Saudi Arabia. Dental assistants, interns, and dental receptionists were excluded. The population of the study was clustered into governmental and private clinics. Then, the subjects were selected within each clinic on a convenience basis with an attempt to have a minimum of five subjects from each site based on usability literature [13,14]. 

The inductive analysis was applied to interview data without any preconceived assumptions. The content of the interviews was transcribed by writing every word, without irrelevant fillers such as “you know,” “ah,” and without including personal information. Thematic analysis was used to analyze the transcripts and included only data based on interactions with patients and the EHR systems [15]. It minimally organizes and describes data as a ground for interview answers [16]. Braun and Clark’s approach was adopted to familiarize with data, generate initial codes, search for themes, review and define themes, and produce the report [16].

## 3. Results 

Thirty-five dentists were interviewed using a convenience sampling approach. The sample was selected from five governmental hospitals and three private dental clinics in Riyadh city, as they met the inclusion criteria. Out of the 35 subjects, 54.3% were male. The average age was 32 (ranged from 25 to 40 years old), and they had practiced dental experiences of 9 years on average (dental experience spanned from 2 to 15 years). Twenty-seven dentists were from dental clinics in governmental hospitals (average of 5.4 from each hospital) and eight were from private hospitals (average of 2.67 from each clinic). All dentists’ offices had computers with an internet connection, and they used emails for clinical purposes such as patient referrals or consultations.

### 3.1. Query for Information 

In all 35 cases, the dentists had inquiries about the chief complaint, COVID-19, the last visit to the clinic, and the last meal, 8% of dentists had inquiries about the medical condition and the patient’s complaint, and 33% had questions about specific medical history. 

The requirements of dentists are different and are based on the patient’s condition, the tools available in the clinic, and the used systems. The number of queries about information (need) was 42 (15 were met and 27 were unmet), with an average of 1.17 per patient. Information needs for the 35 cases were captured through personal interviews.

The outcome of the analysis was aggregated into two categories of information needs: foreground and background information needs.

Query to answer background questions: The inquiries about information were related to general knowledge about health problems. It included the patient’s medical health history, general status, behaviors during previous encounters, insurance coverage, and treatment environment. The dentist’s knowledge of this information helps to better understand the problem’s origin.Query to answer foreground questions: The inquiries about information were related to specific aspects of the current disease: symptoms, procedures, problems, treatment effects, and specific dental health status, in addition to the visual representation of the problem and advanced features such as artificial intelligence capabilities and clinical evidence.

### 3.2. Themes Regarding Dentists’ Information Needs

After we had reviewed all text, we combined all data into groups, called codes (all small encoding). The primary purpose of this activity is to create potential themes that provide help with dentists’ feedback regarding their needs. After analyzing transcripts (Dentist Interviews), we assessed the codes we created to identify patterns and generate the proper themes. The themes were reviewed in terms of the following: Are they present in the data? Did we miss coding anything? Does our analysis cover all text? Can we make our theme better in terms of presentation?

Nine information needs were generated to accurately represent the data collected through dentist interviews to cover all met and unmet needs. We have classified our themes based on foreground and background information needs. Under the foreground information category, chief complaints and symptoms, treatment procedures, medications, visual representation, and advanced features existed as themes. Four themes existed for the background information category: medical and health history, third-party-related insurance coverage, treatment resources, and treatment environment. Table 1 below summarizes all the information needs that dentists reported during the 35 clinical sessions and shows the percentages of cases in which the needs were met or unmet.

### 3.3. Information Needs Related to Chief Complaints and Symptoms

Most of the unmet informational needs were in this category. All dentists who participated in this study reported one or more needs either met or unmet. Thirty-five information needs were raised, twenty-eight where information was unmet, and seven met the dentist’s needs. Thirty-three dentists said they found all information straightforward to access. Accessing patients’ personal information and dental progress notes were the most met information needs.

### 3.4. Information Needs Related to Medical and Health History

Thirty-three dentists reported three information needs: two met and one unmet, and the most met dentist’s need was knowing the patient’s medical history, representing 86%. Similarly, 45.7% were happy to have access to past surgeries. Only 8% of dentists failed to know the patient’s medical history during the treatment session. For example, in the case of Dentist 14, the patient visited the clinic to complete the treatment plan. The dentist said, “He did a treatment plan long ago but did not attend his appointments. He is back now, and I got confused as I found many progress notes forms different dentists. I wish the system would show a clear brief and not FREE text medical history”. Dentist 14 summarized his needs: “I think it would be helpful if the system shows a brief of the medical history instead of going through the progress notes and the X-ray.”

### 3.5. Information Needs Related to Medications

Eleven dentists reported three information needs about prescribed medications, 45%, and allergies representing 36%. Out of 35, 3 dentists had questions about medication. Dentist 4 said, “the purpose of the visit is for a regular dental checkup, and I have asked the patient if he has allergies to any medication and family history…”

### 3.6. Information Needs Related to Visual Representations of the Problem

Twenty-five dentists reported the need for visual representation, and the top need was for the dental chart (graphics and icons), where 68.6% of the dentists asked to have dental chart representation with the ability to add notes on the dental chart and be represented in graphical forms. Dentist 7 said, “… I wish to have dental chart (graphics—icons)…” 

The second need was to have a visual representation in general, where 45.7% of the total dentists wished to have access to any source of visual presentation to help them with better and faster treatment of the problem. For example, one dentist said, “for a patient with a fixed crown, it would be so helpful if the system had an educational feature, like an oral hygiene video instruction and graphics of the dental crown, also having access to search engines.” Another dentist said, “I had a patient earlier who was using a medication with a name I am unfamiliar with (trade name). I needed to search for that medication through my phone (I used web med). It would be helpful if the system could access medical platforms to gather information.” Dentist 18 said, “… it would be helpful if the system highlighted the critical information influencing the dental treatment (blood sugar level, for example).”

### 3.7. Information Needs Related to the Treatment Procedure

Twenty-one dentists reported the need for information for the treatment procedures, and all the information they needed had been met. A total of 90% of the dentists interviewed asked to have treatment plan procedures in the EHR. Dentist 11 was dealing with an extraction problem; he said, “I have looked up in the patient chart, X-ray, and treatment plan and found them all, as it has helped me to understand the problem faster.”

### 3.8. Information Needs Related to X-Ray Services

Twenty-two dentists reported two information needs unmet and one information need met in this category. The top unmet information need was related to access to X-rays. Fifteen dentists (15/22) faced problems requesting X-rays, as some had to exit from EHR and log into another X-ray system. For example, dentist 5 had a restorative treatment session and said, “accessing the X-rays through the current EHR system itself is a hassle.” 

The second issue was with 22.7% of the dentists regarding ordering X-ray images electronically without using manual forms. Three dentists summarized their needs as: “the X-ray is ordered manually, not through the same system.” 

Dentist 18 had a patient come to continue their treatment plan, the dentist tried to order an X-ray and commented, “The current system is too complicated and so not easy to learn, difficult to access the X-rays software from the system itself. The ability to order X-rays from the system is seldom possible. A dental chart with X-rays, when I click on a certain tooth, the X-ray of that specific tooth should pop up…”

### 3.9. Information Needs Related to Advanced Features

The top unmet need was the ability to access external resources through an EHR system, and it was mentioned by 80% of the study’s sample. Twenty-eight dentists mentioned this as an essential need. For example, dentist 7 said, “EHR lacks searching capacity, especially when looking for some information (idea: a pop-up definition window appears once the dentist clicks on a specific word or disease like the window that appears in the online dictionary).” 

The second top-needed function, mentioned by 54% of the study’s sample, was to have messaging features to enhance communication between team members through the EHR system, with 19 dentists stating this. A total of 40% of the dentists asked if there were tutorial videos about how to use an EHR system. 

Another need was to permit the dentist to access the EHR system, which will help them check their appointments and prepare for the next day’s work, as well as a further need which was the ability to edit the notes the day after. Ten dentists revealed that the current system allows one to type notes only at the appointments on the same day. Dentist 4 mentioned, “I cannot edit or add the day after.” Six dentists emphasized the need for a new EHR layout. One dentist said that “the current system has a black background and many text colors regarding the display colors.” 

Three dentists mentioned the need to have a customizable layout. One dentist said, “I wish the system had two modes (like the iPhone), night mode and day mode. Focusing on the patient teeth and long working hours causes eye irritation and dryness and using a system could be discomfiting.” 

On the other hand, three dentists mentioned the need to design the EHR to alert when the next patient arrives. Dentist 3 said, “I wish the system would notify him if the next patient has arrived, as sometimes the dentist is busy with the current patient.”

Of the participants, 54% mentioned the need for advanced features in their EHR. AI-enabled functions, such as speech recognition and natural language processing, represent the system’s ability to process and respond to spoken commands, such as ordering X-rays and medicine. Dentist 1 completed the scheduled treatment for a patient, and he said the following: “Speech recognition, so instead of writing, I can speak, that would save time and effort.”

Thirteen dentists mentioned needs related to education features and question-based history, and four dentists mentioned the need for both features. Dentists mentioned many benefits to having tutorial videos, such as being easy to access anytime, easy to deliver, and effective learning. Tutorial videos would help them to learn whenever they want. 

### 3.10. Information Needs Related to Information about Insurance Coverage

For dentist 29, the session was for an endodontic treatment, and he checked the patient’s medical history and allergies and found all information related to them. He said, “It is essential to know the patient’s insurance and coverage category without referring to any outside source.” Seven dentists were worried if their patients were insured or not and what limitations they might have.

### 3.11. Information Needs Related to the Treatment Environment 

Three information needs were raised by dentists and needed to be met. The first two were related to the dental environment, and the third concerned the patient’s psychological behavior, which was more important than the other two. Is the patient nervous during the treatment session? Or is he calm? Is he afraid of the tools used in the treatment that led to his panic attacks? Knowing all this information helps the dentist determine how to deal with the patient.

Eleven dentists requested this information: 55% asked about dental and psychological behavior, and 36% asked to link the dental laboratory with the tracking process. Dentist 26 emphasized the need for information related to the patient’s behavior. He said: “if there is the possibility to have a psychological behavior-specific window (one time I had a very uncooperative patient, and I wish the previous dentist mentioned that so I can handle it much better).” Further, one dentist was concerned about which treatment environment may positively influence the patient’s experience and said it met his information need.

Table 2 illustrates the top information needs across the above themes, their status, and the percentage of participants who mentioned each need.

## 4. Discussion

The main objective was to collect information about the dentists’ needs for EHR design and functions. However, evaluating the EHR functions is complex, and only some approaches can holistically cover the process. Walji and colleagues recommended multi-methods approaches, including surveys, user testing, and interviews, to assess dentists’ challenges with EHR [17]. Therefore, our study was observational and based on the information provided by the dentists immediately after the treatment session instead of just distributing a questionnaire and gathering information. As all information provided by the dentists was built based on four specific questions without suggesting prior answers, the information needs differed from one dentist to another. Responses to the questions were related to the main complaints and symptoms, treatment options, and procedures.

Dentists envision a dental-oriented EHR to enhance dental care, access the most accurate information, and improve patient outcomes. The EHR was seen to enhance dentists’ ability to plan treatments in one study [18]. Shelley and her colleagues revealed the positive impact of the electronic patient record system on evaluating restorative treatment following root canal therapy [19]. The EHR functions mentioned by dentists in the study emphasize the need for integrated modules for both oral and medical care services. These functions are interrelated, as Atchison and his colleagues mentioned when they assessed 20 years of dental practices and found that only with this integration will dental care excel [20]. Additionally, many shreds of evidence support the need to access a patient’s medical history. One study assessed the impact of electronic dental records (EDRs) on the ability to predict risk factors of osteoradionecrosis of the jaw (ORNJ) in irradiated head and neck cancer patients. Non-dental factors were found to contribute to the ORNJ. This demonstrates the need for an integrated record with complete and accurate dental and medical data [21]. 

Nevertheless, a discrepancy in patient history was declared between parent-reported histories and patient histories recorded in the EHR system. This can seriously compromise patient safety [22]. Therefore, dentists should enhance their medical history skills, which should be learned and optimized in dental education and practice [23]. Medical history contains all the health-related events and is essential in diagnosing and monitoring oral health, especially in the case of multiple sessions. Dentists must refer to various sections in the record, such as hard tissue charts, radiographs, and progress notes. Dentists explain a need for a medical record that contains information about allergies, past surgeries, and previous physical examinations and tests. To reduce inflammation or pain resulting from dental procedures, such as tooth extraction or gum surgery, dentists prescribe medications to relieve this pain and need to explain how long the patient should take the medicine and how to take it. However, dentists need to know about other medications patients take, considering that medications are usually underreported in dental practice [24]. Tenuta and others demonstrated the potential of information technology in enhancing and unifying medication reports in dental settings [22]. Further, triggers can be impeded in dental-oriented medical records to identify and track adverse drug events [25]. 

Generally, dentists would like a system that is easy to use and useful, as it should have functions that cover all dentists’ needs. Further, the usability of the medical record is an essential factor when educating dentists through a proper dental-oriented medical record [24]. When accessing the dental-oriented EHR, the treating dentist must understand and document the patient’s condition and treatment procedure. Dental treatment procedures refer to all services performed during treatment sessions, such as root canals or fillings, or primary services, such as crowns, bridges, and extraction. Usually, the standard dental session time is between 15 min for cleanings with regular dental hygiene and up to 1 h for significant work, depending on the speed and skill of the dentist.

Further, Zanin assessed the literature on the usability of EHRs in dental education and revealed that dentists need a more straightforward way to enter data and use it to analyze the quality of care and for epidemiological purposes [26]. However, using Dental Diagnosis Terminology (DDT) in the current EHRs is seldom supported. One study assessed the introduction of DDT into the EHR and revealed the ease of use in choosing diagnosis terms in the system [27]. Additionally, DDT enhanced dental students’ critical thinking in one study [28]. Access to these specific services is critical in the patient’s treatment journey. X-ray images help dentists to evaluate their patients’ oral health and identify problems such as cavities and tooth decay. Most of the dentists in the study mentioned the inconvenience when requesting X-rays. Dentists must access another system, which is burdensome for dentists because even if they want to go back to the EHR system, they must log out from the X-ray system and then re-enter the EHR to write the medical report. This can be resolved by linking the X-ray system to the EHR. With diagnosis and procedures, dentists in our study recommended capturing data about the treatment environment. This should include tracking patients’ psychological behavior, as it makes treatment sessions easier and enhances patient satisfaction. Therefore, it would be of value to have a mechanism within the dental-oriented EHR to capture information about the patient’s psychological behavior, which reflects the patient’s status during treatment sessions and makes it easy for dentists to manage the treatment session. These recommendations are worth incorporating in the dental-oriented EHR, as the lack of rules, business requirements, and culture infrastructure was demonstrated in the work of Maserat and his colleagues when they conducted a SWOT analysis of electronic dental records [29].

On the other hand, five information needs related to visual representation were raised by dentists. Visual representations are found in the dental chart with an X-ray and dental screening assessment plans. Visual representation helps to better understand the dental problem, with presentations such as pictures, 2D digital images, and 3D visualization in the treatment section. The results of this need agree with the evidence that visualization and dashboards can enhance the ability of dentists to set treatment plans [16]. Further, visualization can enhance instructors’ ability to observe dental students’ performance and using the learning progression dashboard (LPD) has improved students’ overall learning objectives [30]. 

The following needs for dentists in our study were related to having advanced features in the dental-oriented EHR. Many dentists want immediate information to aid them during treatment sessions. Some needs were related to aiding tools for decision-making and artificial intelligence (AI). AI can relieve dentists from routine tasks, aid in diagnosing and setting treatment plans, enable personalized care, and utilize the power of predictive analytics to provide better dental care [31,32]. Using a Dental Decision-Supporting tool can enhance the quality of dental care. One study revealed DDS’s ability to enhance dental students’ critical thinking [28]. Dentronics is a dental robotics system introduced in one dental practice and was able to enhance disease pathogenesis, risk assessment, and disease prediction, and led to better dental care outcomes [33]. However, some issues hinder the broader utilization of AI in dental care as the data quality needs to be substantially enhanced. There is a need to establish a proper, rigorous methodology for assessing the design and development of AI applications. Finally, there are still some unresolved ethical and legal issues around using AI in dentistry [31,32].

The following need of the dentists in our study was related to having access to third-party payers. A link between the EHR and the insurance company helps reduce the processing time and greatly benefits patients by speeding up their treatment. With some restrictions imposed by third-party payers, some dental procedures still need to be fully covered, or some policies are set over simple techniques such as extraction or filling. Therefore, it is essential to know the patient’s insurance coverage to explain to the patient the cost of treatment, the copayments, and the out-of-pocket costs. However, such information is not part of the clinical or dental record as it does not pertain to care. Thus, a separate finance record usually holds insurance benefits, claims, and payment vouchers [34]. 

The dentists in the study mentioned the need to improve the treatment environment, such as keeping hospital spaces clean and safe and fostering a culture of friendliness, which can positively impact the patients’ experience. With the consumerism movement, dental patients expect a more personalized, convenient environment, and this can be attained only by encouraging all healthcare providers to positively interact with patients [20]. 

Nine themes were presented and discussed in our study. To sum up, dentists in our study have provided insights into their needs in the proposed dental-oriented EHR. 

It is worth mentioning that this study has several limitations, as the sample of dentists was chosen on a volunteer basis in one city. The subjects may be systematically different from others who did not participate in this study. Therefore, generalizing the findings of this study to other dentists should be cautiously approached. Second, the nature of this study was exploratory, and future studies can use a more engaging observational approach to verify the results’ validity. Further, it would require allocating more time to conduct an in-depth interview due to the dentists’ packed appointment schedule. More time is needed to allow dentists to elaborate on the answers and investigate their needs more. Finally, the role of allied dental professionals has not been described nor discussed, i.e., the EHR’s needs and functions of dental hygienists, dental assistants, and care coordinators. Thus, their roles and documentation are worth investigating for shedding light on their needs in developing dental-oriented EHR [20]. 

## 5. Conclusions

The need for dental-oriented EHR is well-established. The dentists’ needs in the EHR were recorded through interviews immediately after a treatment session. A thematic analysis was used based on commonalities, and nine themes were described and should be considered when designing the EHR to meet all dentists’ needs. The study’s findings demonstrate gaps between the current and the desired EHR to serve dentists’ needs. For optimal patient outcomes, current EHR systems need better access to patient history and medical information, progress notes, and X-rays, and further need to provide visualization tools for problems and patient charts. Having messaging capability, educational tools, and tutorial videos, and being able to access external resources through an EHR system were among the top unmet needs for dentists in our study. 

## Figures and Tables

**Table 1 healthcare-11-00266-t001:** Themes of information needs.

Need Category	Case with Needs Met	Unmet Need
Chief complaints and symptoms	14	10 (72%)
Visual representations	5	4 (80%)
Advanced features	6	5 (0%)
Treatment procedure	3	0 (0%)
Medications	3	0 (0%)
Treatmtent resources	3	2 (66%)
Medical and health history	3	1 (33%)
Insurance coverage	1	1 (100%)
Treatment environment	4	3 (75%)

**Table 2 healthcare-11-00266-t002:** Specific information needs.

The Top Information Need	Status	Participants (%)
Accessing patients’ personal information and progress notes	Met	94%
Accessing external resources through an EHR system Messaging capability Tutorial videos about how to use an EHR system Accessing X-rays Access to future tasks and appointments The ability to edit the notes Enhance the layout of the EHR Notify the dentist if the next patient has arrived	Unmet	80%
		54%
		40%
		31%
		28.6%
		28.6%
		17%
		8.6%

## Data Availability

Not applicable.

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
