# Peer review of "Informational Needs for Dental-Oriented Electronic Health Records from Dentists’ Perspectives"

_healthcare, 2023, doi:10.3390/healthcare11020266_

Round 1

Reviewer 1 Report

I want to thank the authors for their effort and their article.

1. Language is overall fine.

2. How were dentists selected?

3. I feel more data should be provided regarding these dentists demographic.

4. More information are needed regarding how were these interviews conducted (eg. What questions were asked?)

Author Response

Thank you, I appreciate the opportunity to contribute to Healthcare Journal. 

Attached is a point-to-point response. 

Best regards, 

Reviewer 2 Report

Thanks Author to choose MDPI and Healtcare to publish their manuscript.

the use of EHR is increasingly useful in our clinical practice and allows us to make it much faster and more precise and is a field in vast increase in the field of telemedicine and teleodontology.

It is true that many of the programs used need improvements and modifications which can be observed and corrected from their use, also according to personal opinions.

this also depends on the experience of both digital systems but also on the use that is made. the creation of a multidisciplinary team also with the integration of computer engineers or other sector experts in addition to doctors is necessary.

line 20 typo A.I.

line 129 Braun and ...

Author Response

(The authors gave the same response as above.)

Reviewer 3 Report

This manuscript reports the results of a project concerning the use of electronic health records in dental care. This is a very important topic and the authors are to be congratulated on considering addressing this topic. That said, beyond this very high level statement of the nature of the work, unfortunately it is very unclear precisely what the aim of the project was as well as the methods they used and the findings they report. I will provide some more detailed comments below, but the overall observation is that the authors need to significantly revise the text to make it much clearer to the reader what the goal  of their study is, why they were doing that, what they found and why that is important. Also, there needs to be a much better description of the methods used.

My more detailed comments:

- the introduction section is too long and meandering and so the reader does not know what the rationale of the study is. The high level topic is important but what specifically do the authors want to investigate within the field of EHR and dental care? When they state "the study aims to examine available dental modules.... functions of EHRs......", what is a module? Which functions? An EHR has so many functions, surely they are not examining all? And then "explore the status....." status where, when, in what circumstance?

- Beyond understanding that the authors used qualitative interviews to gather data, we have no idea of the following details:

- who was recruited, why? "dental staff" is mentioned but who does this mean - dentists, dental assistants, receptionists, others?

- there is no description of the setting for the work - the country, is EHR installed, do dentistry use it? the reader has no idea of the context

- what was the sampling strategy, why?

- why 35 participants?

- how were participants approached?

- how did they consent?

- there is no ethics approval recorded

- what was the interview guide? what questions were asked?

- from various comments in the text, it appears that interviews were done with participants as they consulted the EHR with patients - but this is not clear. Were patients present at the time? How did the interview work if participants were checking the EHR at the same time? Were they asked specific questions and then had to look at the EHR to find the answer?

- who did the analyses, how many people, and what training do they have in qualitative interviews, was their triangulation, what did analyzers do with differences in interpretation?

With respect to the results, I will take one section as an illustration of the difficulty of understanding them. Given we don't know the study aims and the methods and the context of the study are very unclear, this already makes it very difficult to understand the results. If we take the example of the theme of "information needs":

- it is not clear what the background, foreground and general questions/information are? from the description provided I think they are referring to EHR content ie the dentists would like to be able to obtain certain information on each patient when looking at their EHR. But in Table 1 what does "case with needs" mean? And what does "EHR functions" mean? and how can that be met or unmet?

- In table 1 what does "AI features" mean? The text apparently related refers to "Advanced features" but then goes back to discussing AI functions - so is the study also investigating mechanisms and processes not just content of the EHR?

- the text concerning "needs related to the medical and health history" contains some percentages (which are not appropriate in qualitative research) but also cites some quotations (which is appropriate) but it is not at all clear what point is being made. Dentist 14 seems to be referring to a particular case - were all participants shown a case? If so the same case? And what questions were asked of this case? Is dentist 14 describing what s/he reads in an EHR? Is he an example of a dentist who "failed to know the patient's medical history"?

In summary, there are many elements of the text that are very unclear. I have read the article several times and remain unclear what was done and what the findings were?

Author Response

(The authors gave the same response as above.)

Reviewer 4 Report

The topic of the manuscript is to examine the available dental modules and the functions of electronic health records from the perspectives of dental care providers.

The title and the abstract of the article are informative. The Introduction relatively precisely presents the issue of Electronic Health Record in dentistry. The section "Material and Methods" very briefly describes the chosen study design. The section "Results" requires some revisions. The Discussion is interestingly written, however, the paragraph about the study limitations should be expanded. The Conclusions should be revised.

Some following points must be clarified/corrected for the further processing of this article.

Merits-related comments:

1.       The purpose of the study should be made more specific instead of a few sentences.

2.       The Materials and Methods section needs to be significantly expanded - how participants were selected, who conducted the interview, what about calibration and validation etc.

3.       In supplementary materials, please add the questionnaire.

4.       It would be worth considering whether needs differed by gender or workplace.

5.       In Table 1, it is not worth duplicating the column "unmet", when we have the sum of cases and the column "met".

6.       The presentation of results (e. g., Table 2) lacks some consistency. Once about "met," once about "unmet" - please consider how to organise this section to make it reader-friendly

7.       The study limitations should be expanded at the end of the Discussion.

8.       The Conclusions should be modified to be “take-home” messages regarding the study findings.

Technical comments:

1.       The manuscript requires editorial editing, e. g. punctuation.

2.       The abstract should be a single paragraph and should follow the style of structured abstracts, but without headings.

3.       References should be described as follows:
1. Author 1, A.B.; Author 2, C.D. Title of the article. 
Abbreviated Journal Name YearVolume, page range.

Author Response

(The authors gave the same response as above.)

Round 2

Reviewer 4 Report

The Authors have addressed most of the comments, improving the manuscript. However, there is still no appropriate style of the references in accordance with the guidelines of the journal.

Author Response

Dear Reviewer,

Thank you very much for taking the time to review our manuscript. We tried to fix the reference style to meet the required journal style.

Thank you again.